# Waymax: An Accelerated, Data-Driven Simulator for Large-Scale Autonomous Driving Research

**Cole Gulino** [*†]     **Justin Fu** [*†]     **Wenjie Luo** [*†]     **George Tucker** [*‡]     **Eli Bronstein** [†]

**Yiren Lu** [†]   **Jean Harb** [†]   **Xinlei Pan** [†]   **Yan Wang** [†]   **Xiangyu Chen** [†]   **John D Co-Reyes** [‡]

**Rishabh Agarwal** [‡]     **Rebecca Roelofs** [‡]     **Yao Lu** [‡]     **Nico Montali** [†]     **Paul Mougin** [†]

**Zoey Yang** [†]       **Brandyn White** [†]       **Aleksandra Faust** [‡]       **Rowan McAllister** [†]

**Dragomir Anguelov** [†]                    **Benjamin Sapp** [†]

* Equal Contribution        [†] Waymo Research        [‡] Google DeepMind

## Abstract

Simulation is an essential tool to develop and benchmark autonomous vehicle planning software in a safe and cost-effective manner. However, realistic simulation requires accurate modeling of nuanced and complex multi-agent interactive behaviors. To address these challenges, we introduce Waymax, a new data-driven simulator for autonomous driving in multi-agent scenes, designed for large-scale simulation and testing. Waymax uses publicly-released, real-world driving data (e.g., the Waymo Open Motion Dataset [15]) to initialize or play back a diverse set of multi-agent simulated scenarios. It runs entirely on hardware accelerators such as TPUs/GPUs and supports in-graph simulation for training, making it suitable for modern large-scale, distributed machine learning workflows. To support online training and evaluation, Waymax includes several learned and hard-coded behavior models that allow for realistic interaction within simulation. To supplement Waymax, we benchmark a suite of popular imitation and reinforcement learning algorithms with ablation studies on different design decisions, where we highlight the effectiveness of routes as guidance for planning agents and the ability of RL to overfit against simulated agents.

## 1   Introduction

Due to the cost and risk of deploying autonomous vehicles (AVs) in the real world, simulation is a crucial tool in the research and development of autonomous driving software. The two primary challenges of a simulator are speed and realism: we wish for a simulator to be fast in order to cost-effectively train/evaluate on many hours of synthetic driving experience, and we wish for a simulator to be diverse and realistic in terms vehicle behavior in order to minimize the sim-to-real gap [50, 37], such that performance in the simulator correlates with real-world performance.

Existing work in simulation for autonomous driving has made significant progress in recent years. Simulators such as CARLA [14], Sim4CV [33] and SUMMIT [9] focus on photo-realistic rendering of driving scenarios, enabling users to train and evaluate driving solutions. However, a major

37th Conference on Neural Information Processing Systems (NeurIPS 2023) Track on Datasets and Benchmarks.

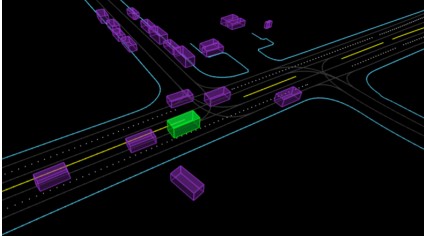 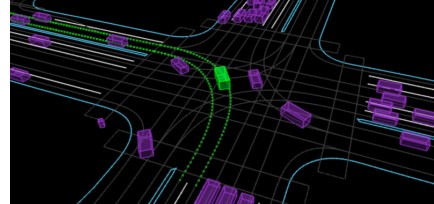

(a) Waiting for a turn into oncoming traffic.            (b) Navigating a 4-way intersection.

Figure 1: Two examples demonstrating the types of interactive, urban driving scenarios available in Waymax. **(a)** shows a vehicle waiting for oncoming traffic to pass before turning into a narrow street. **(b)** shows an agent performing an left turn at a 4-way intersection while following a route (boundaries highlighted in green).

simulation challenge still remains in the generation of diverse scenarios and realistic behavior for other agents (such as vehicles and pedestrians) in the scene, and as the driving field has matured, *behavior challenges* have been shown to be a significant bottleneck to scaling [31]. To this end, there is still a need for simulation tools that provide (a) realistic, closed-loop simulation of agent behavior, and (b) high speed and throughput to support modern trends in machine learning that use large models and datasets.

To address these challenges, we propose Waymax- a *differentiable*, *hardware-accelerated* and *multi-agent* simulator that is built using *real-world driving data* from the Waymo Open Dataset. Waymax aims to provide, within simulation, a faithful reproduction of the data and types of challenges a real autonomous driving agent would face, such as those shown in Fig. 1. Waymax simulates challenging obstacles present in urban driving, such as pedestrians and cyclists, and provides high-level route information for the ego vehicle to follow. To optimize runtime speed and facilitate rapid development, Waymax is written using JAX [5], which allows simulation to be run entirely on accelerators such as graphics and tensor processing units (GPUs and TPUs). To provide better simulation realism, Waymax uses diverse scenarios initialized from the Waymo Open Motion Dataset (WOMD) [15], which contains over 250 hours of real driving data collected in dense urban environments. Waymax data loading and processing can be extended to other popular datasets without loss of generality.

Our contributions are two-fold. First, we introduce the Waymax simulator, which is a multi-agent simulator for autonomous driving that is (a) hardware-accelerated, (b) provides features and routes information from real driving data, and (c) constructs scenarios upon a large and diverse dataset of real-world driving. Our second contribution is in providing a set of common benchmarks and simulated agents that allow researchers to score and benchmark their autonomous planning methods in closed-loop. We show-case the flexibility of Waymax by training behavior algorithms for an autonomous vehicle in different setups (imitation, on and off policy RL, etc) against a range of different interactive agents.

## 2   Related Work

**Simulators for Autonomous Driving.**   Waymax is a *differentiable*, *hardware-accelerated* and *multi-agent* simulator that is built on top of *real-world driving data*. We compare our work to other related publicly available simulators in Table 1. The closest works to ours are multi-agent autonomous driving simulators which use real driving data to initialize scenarios and logged behavior, such as Nocturne [52], MetaDrive [25], and nuPlan [8]. In comparison, our simulator is designed to support hardware-accelerated training, the agent models can be connected in-graph for training and inference, and the simulation is differentiable. We provide a full set of features including pedestrians, cyclists, and traffic lights available in WOMD, and the inferred routes for goal-conditioned policy and progress metrics. Additionally reactive sim agent models are provided to facilitate realistic simulation. TorchDriveSim [46] is the only public simulator that supports differentiable simulation for in-graph acceleration. In comparison, we provide rich and diverse real-world human expert driving data from WOMD. It is worth noting that the driving policy modeling problem is primarily focused on behavior than perception, thereby, we do not intend to support sensor simulation, such as [14, 28, 48, 9, 2], which represents another important research area for autonomous driving perception.

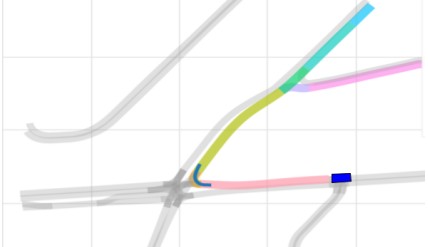
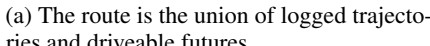
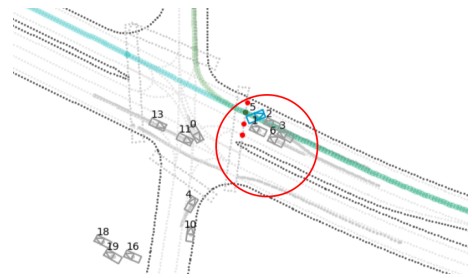

(a) The route is the union of logged trajectories and driveable futures.

(b) Reactive simulated agents stopping to avoid collision.

Figure 2: A sample of features available in Waymax. a): The routes given to an agent (all areas highlighted in color) are computed by combining the logged future trajectory of the agent with all possible future routes after the logged trajectory. b): Waymax is bundled with reactive simulated agents. Here, agent #5 (circled in red) is stopped in front of an intersection, causing the IDM-controlled agents (#1, 2, 3, and 6) to brake in order to avoid collision.

| | Multi-agent | Accel. | Sensor Sim | Expert Data | Sim-agents | Real Data | Routes/Goals |
|---|---|---|---|---|---|---|---|
| TORCS [55] | | | ✓ | | ✓ | | - |
| GTA V [29] | | | ✓ | | | | - |
| CARLA [14] | | | ✓ | | ✓ | | Waypoints |
| Highway-env [24] | | | | | | | - |
| Sim4CV [33] | | | ✓ | | | | Directions |
| SUMMIT [9] | ✓(≥ 400) | | ✓ | | ✓ | ✓ | - |
| MACAD [38] | ✓ | | ✓ | | ✓ | | Goal point |
| DeepDrive-Zero [40] | ✓ | | | | ✓ | | - |
| SMARTS [57] | ✓ | | | | | | Waypoints |
| MADRaS [44] | ✓(≥ 10) | | ✓ | | ✓ | | Goal point |
| DriverGym [23] | | | | ✓ | ✓ | ✓ | - |
| VISTA [2] | ✓ | | ✓ | ✓ | | | - |
| nuPlan [8] | | | ✓ | ✓ | ✓ | ✓ | Waypoints |
| Nocturne [52] | ✓(≥ 50) | | | ✓ | ✓ | ✓ | Goal point |
| MetaDrive [25] | ✓ | | ✓ | ✓ | ✓ | ✓ | - |
| Intersim [47] | ✓ | | | ✓ | ✓ | ✓ | Goal point |
| TorchDriveSim [46] | ✓ | ✓ | | | ✓ | | - |
| tbsim [56] | ✓ | | | ✓ | ✓ | ✓ | Goal point |
| Waymax (ours) | ✓(≥ 128) | ✓ | | ✓ | ✓ | ✓ | Waypoints |

Table 1: A comparison of related driving simulators (chronological order). *Multi-agent.*: Simulating multiple agents. Supported number of agents are in the parentheses (if available). *Accel.*: In-graph compilation for hardware (GPU/TPU) acceleration. *Sensor Sim*: Sensors (e.g. camera, lidar & radar) input simulation. *Expert Data*: Human demonstrations or rollout trajectories collected with an expert policy. *Sim-agents*: agent models for simulated objects (e.g. other vehicles). *Real data*: Real world driving data. *Routes/Goals*: "−" means no routes or goals are provided; "Waypoints" means positions sampled from a trajectory; "Directions" means discrete driving directions including left, straight, and right; "Goal point" means the goal position.

**Learning-based Driving Agents.** In the context of autonomous driving, open-loop imitation learning (IL), known as behavior cloning (BC), has been applied to predict driving behaviors of other road users [3, 10, 49, 26, 36, 17, 34] as well as the ego vehicle [39, 4, 41, 12, 53, 11]. It is widely known that BC suffers from covariate shift [42] and causal confusion [13]. Closed-loop methods including adversarial imitation learning [20, 7] and reinforcement learning (RL) methods [22, 21, 54, 27] have been proposed to address these challenges by learning from feedback and explicit hand designed rewards in the simulator. Despite the excitement of machine learning as an avenue towards devising autonomous driving policies, benchmarking different learned policies on large scale real world datasets remains a challenge. To enable accelerated and effective autonomous driving agents research, Waymax enables standard training and evaluation workflows and reliable benchmarking in both open

and closed-loop settings; we also provide the implementation of a representative set of IL and RL baselines and report their performance against a standard set of metrics on Waymax as references.

# 3 Simulator Features

In this section, we give an overview of the features of Waymax and its interface to the user. Waymax is a simulator that supports controlling arbitrary number of objects in a scene. A primary goal of Waymax is to initialize from real-world driving scenarios to model complex interactions between vehicles, pedestrians, and traffic lights, and following a goal or route provided by high-level planner. Additionally, Waymax is designed to be both fast and flexible - each component discussed in this section can easily be modified or replaced by an user to suit their own project needs. We discuss the scenarios and datasets in Sec. 3.1, state representation in Sec. 3.2, and the dynamics and action representation in Sec. 3.3. Waymax includes a suite of common metrics described in Sec. 3.4, and several options for modeling the behavior of dynamic objects (vehicles and pedestrians) in the scene, outlined in Sec. 3.5.

## 3.1 Scenarios and Datasets

In contrast to simulators that generate synthetic scenarios (e.g. CARLA [14]), Waymax utilizes real-world driving logs to instantiate driving scenarios, and runs for a fixed number of steps. We provide default support for the Waymo Open Motion Dataset (WOMD) [15], which includes over $100,000$ trajectories snippets and 7.64 million unique objects to interact or control. Each trajectory snippet is 9 seconds recorded with 0.1 Hz. The trajectory contains pose and velocity information for all objects in a scene, including the autonomous vehicle (AV), other vehicles, pedestrians, and cyclists. For each scenario, we take the static information such as the road graph and initialize dynamic objects using the first second of logged information. Then, agent models (described in Sec. 3.5) will be used to control the dynamic objects such as pedestrians and the other vehicles through the simulation steps. Note importantly, users can inject multiple agent models and dynamics model to Waymax environment where each model can control multiple objects.

## 3.2 State and Observation spaces

The first component of defining autonomous driving as a sequential control problem is defining the state space. We include two types of data in the state: **dynamic** data which can change over the course of an episode and across scenarios, and **static** data which remains the same during an episode but varies across scenarios. The dynamic data in the state consists of the position, rotation, velocity, and bounding box dimensions for all vehicles, cyclists, and pedestrians in a scene, along with the color of traffic light signals (red, yellow, green). The static data includes the road and lane boundaries sampled as a 3D point-cloud (known as the "roadgraph"), as well as on-route and off-route paths for the ego vehicle. Each agent views the simulator state through a user-defined observation function, which can induce *partial observability*. We provide a default observation function that transforms the location of all other vehicles to the agent's own coordinate frame, and sub-samples the roadgraph via distance.

**On-Route and Off-Route Paths** We augment each scenario with feasible paths that the AV could take from its initial position. A path is represented as a sequence of points, which are a subset of the roadgraph points. Each path is computed by performing a depth-first-search traversal of the roadgraph from the starting position. Together, these paths describe all the ways in which the AV can legally drive in the scenario. Similar to the "road-route" in [7], a path is considered on-route if it follows the same road as the AV's logged trajectory. The remainder of the paths that are not on-route are deemed to be off-route. Fig. 2a gives an example of on-route paths. These paths are useful for computing metrics as well as developing goal-conditioned planning and interactive agents.

## 3.3 Object Dynamics

The object dynamics defines what 'actions' an object would expect and how its state would evolve given an action. Waymax allows the user to define a dynamics model and provides several pre-defined options for controlling the physical dynamics of vehicles in simulation: 1) the *delta action* space,

which is suitable for all types of objects, uses position difference (the delta term $\Delta x, \Delta y, \Delta \theta$)) between two consecutive states; and the *bicycle* action space (($a, \kappa$), which is only for vehicles, uses acceleration and steering curvature). The equations defining these dynamics can be found in Appendix A.1.

## 3.4 Metrics

Waymax provides a set of intuitive metrics to evaluate the ego vehicle as well as simulated agents for safety and correctness of behavior (such as obeying traffic rules, not colliding), as well as comfort and progress. All metrics in Waymax are computed in *closed-loop*, meaning that they are computed by running the agent in simulation, rather than in *open-loop*, where metrics are computed on a per-timestep basis without feedback from simulation. The metrics available are as follows:

**Route Progress Ratio** The route progress ratio measures how far the ego vehicle drives along the goal route compared to the logged trajectory. At time step $t$, this metric associates the vehicle's position to the closest point $x(t)$ in an *on-route* path. It then computes the distance along the path from the start of the path to $x(t)$, denoted as $d_{x(t)}$. The route progress ratio is then defined as $\frac{d_{x(t)} - d_p}{d_q - d_p}$, where $d_p$ and $d_q$ are the distances along the path to the initial and final positions of the vehicle's logged trajectory, respectively. Since the vehicle can continue driving after reaching its destination, this ratio could be greater than 1.

**Off-Route** The off-route metric is a binary value indicating if the vehicle is following an on-route path. If the vehicle is sufficiently closer to an off-route path than on-route path or it is far enough away from an on-route path, it is considered off-route.

**Off-Road** The off-road metric triggers if a vehicle drives off the road. This is measured relative to the oriented roadgraph points. If a vehicle is on the left side of an oriented road edge, it is considered on the road; otherwise the vehicle is considered off-road.

**Collision** The collision metric is a binary metric that measures if the vehicle is in collision with another object in the scene. For each pair of objects, if the 2D top-down view of their bounding boxes overlap in the same timestep, they are considered in collision.

**Kinematic Infeasibility Metric** The kinematic infeasibility metric computes a binary value of whether a transition is kinematically feasible for the vehicle. Given two consecutive states, we first estimate the acceleration and steering curvature using the inverse kinematics defined in Appendix A.1, and check if the values are out of bounds. We empirically set the limit of acceleration magnitude to be $6 \ m/s^2$ and the steering curvature magnitude to be $0.3 \ m^{-1}$. In order to determine these empirically, we fit the logged trajectories of the ego agent with our steering and acceleration action space. We then chose the limits to be roughly the maximum (rounding up for some slack) of the values we observed in the logs.

**Displacement Error** The average displacement error metric (ADE) measures how far the simulation deviates from logged behavior. It is defined as the L2 distance between each object's current XY position and the corresponding position recorded in the logs at the current timestep, averaged across all timesteps.

## 3.5 Simulated Agent Behavior

An important part of constructing a simulator for autonomous driving is realistic behavior for simulated agents other than the AV. Waymax, as a multi-agent simulator, gives the user the ability to control the behavior of all objects in simulation. This allows the user to control agents with any model of choice, such as learned behavior models. However, to support training AV agents out-of-the-box, Waymax also includes a rule-based reactive agent model based on the intelligent driver model (IDM) [51]. IDM describes a rule for updating the acceleration of a vehicle to avoid collisions based on the proximity and relative velocity of the vehicle to the object directly in front of the vehicle, as demonstrated in Fig. 2b. The IDM agent in Waymax follows the logged path that is

recorded in the data, but uses IDM to adjust the speed profile to avoid collisions and accelerate on free roads.

## 4 Software API

We now outline the Waymax software components and interfaces. In order to support a wide variety of research workflows, Waymax is designed as a collection of inter-operable libraries while maintaining fast simulation speed. The main libraries comprise of (1) a set of common data-structures, (2) a distributed data-loading library, (3) simulator components such as metrics and dynamics, and (4) a Gym-like environment interface. Each component of the simulator can be modified, replaced, or used standalone by the user. In this manner, users who only need one component of Waymax (e.g. only metrics, or only data loading), or who wish to significantly modify the behavior of the simulator (such as generating synthetic scenerios) can easily do so through Waymax's APIs.

### 4.1 Environment Interface

Users primarily interact with Waymax as a partially-observable stochastic game. The Waymax interface follows the the Brax [16] design to only define functionally pure initialization and transition functions. This stateless design enables efficient optimization through JAX's [5] JIT compiler and functional libraries, and easily allows users to implement control algorithms that require backtracking, such as search. In contrast with stateful simulators, such as OpenAI Gym [6] and DM Control [32], Waymax users need to maintain the simulator state within a simulation loop and interact with the simulator primarily through two functions:

- The reset(scenario) function takes as input a raw scenario, performs any initialization necessary such as populating the simulation history, and returns the initial state object.
- The step(state, action) function takes as input the current state, the actions for all agents, and computes the successor state as well as the new observation and metrics. The actions argument is a data structure that contains a data tensor of actions for each agent, as well as a validity mask which denotes which agents the user wishes to control. step then returns these results in a new timestep object.

Using these two functions, a user can run a simple, but complete simulation of a stochastic game between multiple agents, such as in the following pseudocode example:

In addition, we do provide adapters to convert the functionally pure Waymax simulator into a stateful one to support existing codebases.

```
# Run one episode until termination.
state = env.reset(next(dataset))
while not done:
  action = policy(env.observe(state))
  state = env.step(state, action)
```

### 4.2 Hardware Acceleration and In-graph training.

Waymax supports both hardware acceleration on GPUs and TPUs, as well as combining training and simulation within the same computation graph (referred to as "in-graph" training), which allows training and simulation to happen entirely on the accelerator without communication bottlenecks through the host machine. These features are possible because Waymax is written entirely using the JAX [5] library, which converts operations into XLA [43], a linear algebra instruction set and optimizing compiler which supports execution on CPU, GPU, or TPU. In-graph training requires the modeling and training code to be written using an XLA-compatible frontend such as JAX [5], or Tensorflow [1]. The XLA compiler can then optimize combined training and simulation program to produce a single computation graph that can be run entirely on hardware accelerators, without communication costs between the accelerator and host device.

### 4.3 Single and Multi-agent Simulation

While the base multi-agent environment allows us to do sim-agents (multi-agent) learning similar as in Nocturne [52], MetaDrive [25], the ultimate goal of the autonomous driving problem is to train an AV planning agent. Thus, Waymax supports both multi-agent simulation that allows users to control

| | Device | BS-1 | BS-16 | Reset | Step | Transition | Metrics | RolloutExpert |
|---|---|---|---|---|---|---|---|---|
| Single-Agent Env | CPU | ✓ | | 1.09 | 131 | 0.90 | 112 | $1.0\times10^4$ |
| | CPU | | ✓ | 12.2 | $1.7\times10^3$ | 10.9 | $1.69\times10^3$ | $1.4\times10^5$ |
| | GPU-v100 | ✓ | | 0.58 | 0.75 | 0.47 | 0.21 | 56.2 |
| | GPU-v100 | | ✓ | 0.67 | 2.48 | 0.52 | 2.27 | 279 |
| Multi-Agent Env | CPU | ✓ | | 6.23 | 129 | 1.01 | 112 | $1.1\times10^4$ |
| | CPU | | ✓ | 49.8 | $1.1\times10^3$ | 14.3 | $1.72\times10^3$ | $1.6\times10^5$ |
| | GPU-v100 | ✓ | | 0.64 | 0.92 | 0.53 | 0.19 | 73.3 |
| | GPU-v100 | | ✓ | 0.81 | 2.86 | 0.51 | 2.24 | OOM |

Table 2: Runtime benchmark in milliseconds: the environment controls all objects in the scene (up to 128 as defined in WOD).

arbitrary objects within the scenario, as well as a single-agent workflow where a single AV agent is trained using learned or rule-based models to control the other vehicles in the scene.

While it might be possible to put multiple policies in one environment directly, it is certainly not a flexible way as it is hard to coordinate different policies or change policies. Waymax provides two interfaces for different use-cases:

The **MultiAgentEnvironment** provides an interface for multi-agent and sim-agent problems. The user provides simultaneous actions for all controlled objects in the scene, as well as a mask to indicate which objects should be controlled.

The **PlanningAgentEnvironment** exposes an interface for controlling only the ego vehicle in the scene. All other agents are controlled by user-specified sim agents or log playback (Fig. 3).

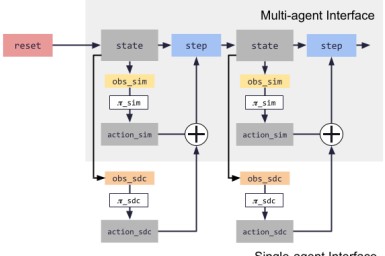

Figure 3: An illustration of a simulation rollout using reactive simulated agents to control non-AV agents, and a user-defined policy to control the AV.

## 5 Experiments

We now evaluate both Waymax as a simulator and the performance of several reference agents simulated using Waymax. We first evaluate the computational performance of Waymax in Sec. 5.1 under various configurations. Second, we perform an empirical study of several benchmark agents for planning in Sec. 5.3, where we compare the performance of several broad categories of learned planning algorithms (such as imitation learning and reinforcement learning) against both logged agents and reactive simulated agents. For the second part, our goals was to showcase potential options for using Waymax, so we opted for simple design choices and a breadth of configurations, and we expect that the performance of the baseline agents could be significantly improved in future work.

### 5.1 Runtime Benchmark

In Tab. 2, we present the runtime performance of Waymax using a CPU (Intel Xeon W-2135@3.7GHz) and a GPU (Nvidia-V100). We evaluate the performance of both multi-agent and the single-agent environment with different batch size. All functions are jit compiled and runtime is reported in millisecond. Following WOMD, the environment controls up to 128 objects in one scene. Note that the Step function computes both the state transition and the reward. While users specify customized reward function, for this runtime evaluation, we use the negative sum of all metrics in 3.4 as the reward, which measures the effect of computing all metrics. When considering batch size 1 and using a GPU, Waymax achieves over 1000Hz for Step function and over 2000Hz if only considering the Transition. More importantly, as Waymax supports batching, Step only takes 2.86ms using a batch size of 16. Note this is much faster than running batch size one for 16 times and gives an equivalent runtime of over 5000Hz per example (i.e. closer to 500 times faster than using a CPU). Noticeably the Metrics function consumes more computation then the Transition function because the *Off-Road* metric needs to find nearby roadgraph points, which is a slow operation.

| Agent | Action Space | Train Sim Agent | Off-Road Rate (%) | Collision Rate (%) | Kinematic Infeasibility (%) | Log ADE (m) | Route Progress Ratio (%) |
|---|---|---|---|---|---|---|---|
| Expert | Delta | - | 0.32 | 0.61 | 4.33 | 0.00 | 100.00 |
| Expert | Bicycle | - | 0.34 | 0.62 | 0.00 | 0.04 | 100.00 |
| Expert | Bicycle (Discrete) | - | 0.41 | 0.67 | 0.00 | 0.09 | 100.00 |
| Wayformer | Delta | - | 7.89 | 10.68 | 5.40 | 2.38 | 123.58 |
| BC | Delta | - | 4.14±2.04 | 5.83±1.09 | 0.18±0.16 | 6.28±1.93 | 79.58±24.98 |
| BC | Delta (Discrete) | - | 4.42±0.19 | 5.97±0.10 | 66.25±0.22 | 2.98±0.06 | 98.82±3.46 |
| BC | Bicycle | - | 13.59±12.71 | 11.20±5.34 | 0.00±0.00 | 3.60±1.11 | 137.11±33.78 |
| BC | Bicycle (Discrete) | - | **1.11±0.20** | **4.59±0.06** | 0.00±0.00 | **2.26±0.02** | 129.84±0.98 |
| DQN | Bicycle (Discrete) | IDM | 3.74±0.90 | 6.50±0.31 | 0.00±0.00 | 9.83±0.48 | 177.91±5.67 |
| DQN | Bicycle (Discrete) | Playback | 4.31±1.09 | 4.91±0.70 | 0.00±0.00 | 10.74±0.53 | 215.26±38.20 |

Table 3: Baseline agent performance evaluated against IDM sim agents with route conditioning. Models trained ourselves (BC and DQN) report mean and standard deviation over 3 seeds. Off-Road, Collision, and Kinematic Infeasibility are reported as a percentage of episodes where the metric is flagged at any timestep. Action spaces are continuous unless noted otherwise. By construction the bicycle action space does not violate the comfort metric.

**Rollout**   We also benchmark a `Rollout` function which rolls out the environment given an *Actor* for an entire episode (i.e., 80 steps for WOD). This is especially useful to provide faster inference and evaluation. In the last column of Tab. 2, we show the runtime of `Rollout` with an `ExpertActor` that derives grouth-truth actions from logged trajectory. It is faster than running `Step` function 80 times. More importantly, we can see that running on GPU has a consistent 2 orders of magnitude speedup. As a point of reference, evaluating the full WOD evaluation dataset (44K scenarios) with 8-V100 machine takes less than 2min.

## 5.2   Baseline Planning Agents

**Expert**   We provide a number of expert agent models to provide groundtruth actions for open-loop training. Each agent uses the inverse function of the action spaces defined in Section 3.3 to fit an action to the logged trajectory. For discrete action spaces, the inverse is computed by discretizing the continuous inverse.

**Behavior Prediction Model (Wayformer)**   As a point of reference, we adapt the state-of-the-art Wayformer behavior prediction model [35] to the planning setting. Originally, the Wayformer predicts multiple 8-second future trajectories given a 1-second context history. To adapt it to the planning setting, we autoregressively feed in its predictions as the context history and choose the most likely trajectory. We found that making predictions at a lower frequency than the environment frequency improved performance, so we predict 5-step long trajectories and only replan every 5 steps.

**Behavior Cloning**   We re-use the encoder portion of Wayformer [35] followed by a 4-layer residual MLP to maximize the log likelihood of the expert actions. For continuous actions, we used a 6-component Gaussian Mixture Model. For discrete actions, we used a softmax layer to compute action probabilities.

**Model-Free Reinforcement Learning - DQN**   We used the Acme [19] implementation of prioritized replay double DQN [45].

We used the same architecture as in discrete BC for the Q-network, interpreting the logits of the model as Q-values.

For simplicity, we use a sparse reward penalizing collisions and off-road events: $r_t = -\mathbb{I}_{\text{collision}}(t) - \mathbb{I}_{\text{off-road}}(t)$.

## 5.3 Planning Benchmark Results

To showcase the flexibility of our environment, we trained a number of baselines on different action spaces and algorithms as shown in Table 3 and evaluated the metrics defined in Section 3.4. We evaluated each agent against the IDM sim agent and conditioned it on the route by adding the points from all the on-route paths as an additional input group to the Wayformer [35] encoder. These points represent the on-route subset of the roadgraph points. All agents are trained for the planning agent task and thus only provide predictions for the autonomous vehicle. See Appendix A.2 for training details.

As expected, the expert agents have low off-road and collision rates. The nominal values represent noise in the bounding boxes and logged data and serve as a lower bound for performance. The expert using the discrete bicycle action space has comparable performance to the other experts, confirming that the discretization is sufficiently fine.

For open-loop imitation, the discrete action space performs best, possibly because it is easier to model multi-modal behavior. Furthermore, it outperforms the adapted Wayformer model, likely due to the fact that it is trained explicitly for this task. This serves as a check that the Waymax environment is producing the correct training data.

**Route Conditioning Ablation** To showcase the utility of route conditioning, we compare the performance of route conditioned versus non-route conditioned behavior cloning agents Table 4 shows that the route conditioned agent is substantially better at following the route, while also achieving a lower off-road rate, collision rate, and log ADE. These results indicate that the route provides a strong signal for the planning task.

| Agent (Action Space) | Off-Road Rate (%) | Collision Rate (%) | Off-Route Rate (%) | Log ADE (m) | Route Progress Ratio (%) |
|---|---|---|---|---|---|
| Expert (Bicycle Discrete) | 0.41 | 0.67 | 0.00 | 0.00 | 100.00 |
| BC (Bicycle Discrete) | 1.45±0.05 | 4.92±0.24 | 2.31±0.12 | 2.41±0.06 | 128.73±1.50 |
| BC (Bicycle Discrete) + Route | **1.11±0.20** | **4.59±0.06** | **0.96±0.09** | **2.26±0.02** | 129.84±0.98 |

Table 4: Experimental ablation comparing performance with and without route conditioning.

**Sim Agent Ablation** In Table 5, we show the effect of training and evaluating an imitation agent against IDM sim agents versus playing back logged trajectories. As expected, evaluating with the IDM agent produces fewer collisions than evaluating with log playback. However, training an RL agent with IDM agents was less effective than training against logged agents. We believe this is because the RL agent tends to overfit or exploit the behavior of 'easier' IDM agents. Since IDM will stop for the SDC to avoid collisions, the RL agent does not have as much incentive to learn how to avoid collisions itself. We can see that when an IDM-trained agent is evaluated against logged agents, the collision rate is over 4x higher than when evaluated against IDM agents.

| Train Agent | Eval Agent | Collision Rate (%) | Offroad Rate (%) | Progress Ratio (%) |
|---|---|---|---|---|
| Playback | Playback | 8.67±0.97 | 5.16±1.00 | 193.56±26.82 |
| Playback | IDM | **4.91±0.70** | **4.31±1.09** | 215.26±38.20 |
| IDM | Playback | 25.15±0.76 | 4.91±0.41 | 163.11±4.51 |
| IDM | IDM | 6.50±0.31 | **3.74±0.90** | 177.91±5.67 |

Table 5: Experimental ablation over different configurations of train/evaluation sim agents with the DQN algorithm using the Bicycle dynamics model. Using interactive sim agents such as IDM can reduce the rate of unrealistic collisions during evaluation. However, it is more difficult to train effective planning agents using interactive sim agents.

# 6 Conclusion

We have presented Waymax, a multi-agent simulator for autonomous driving. Waymax provides diverse scenarios drawn from real driving data, and supports hardware acceleration and distributed training for efficient and cost-effective training of machine-learned models. It is also designed with flexibility in mind - Waymax is written as a collection of inter-operable libraries for data loading, metric computation, and simulation, which can support a wide variety of research problems that are not limited to just the planning evaluations presented in this work. We conclude by benchmarking several common approaches to planning with ablation studies over different dynamics and action representations, which provide a set of strong baselines for benchmarking future work.

In addition to hardware acceleration, Waymax also enables the exploration of methods utilizing differentiable simulation, as the entire simulation can be assembled within a single JAX computation graph. Prior work[30, 20] has shown that differentiable simulation can improve the efficiency of policy optimization methods as they can rely on a "reparameterized" or pass-through gradient to reduce the variance of the gradient estimate. We believe that this is a promising line of future work to be explored.

As mentioned previously, the problem of sim-to-real transfer is a critical issue in autonomous driving, as it is cheap and desirable to evaluate in simulation but difficult to guarantee that the same performance and level of safety will carry over to the real world. While in Waymax we have made design decisions to minimize this gap (such as using real-world data to seed scenarios), this remains an important limitation for any simulation-based framework. A fruitful line of future work is to close the gap between simulated and real-world performance, potentially using techniques such as domain randomization [50] or combining real and synthetic data[37, 18].

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
