# A  Appendix

## A.1  Dynamics Definitions

**Delta Action Space**. Define an agent's current state information as $s = (x, y, \theta, v_x, v_y)$, which includes the $x, y$ positions in the coordinate space, and the yaw angle $\theta$, and the velocities in the $X$ and $Y$ directions. Given the action $(\Delta x, \Delta y, \Delta \theta)$, which accounts for the change in the positions and yaw angle of the agent, and given the time step length for one step $\Delta t$, the next state $s' = (x', y', \theta', v'_x, v'_y)$ can be expressed as,

$$
\begin{aligned}
x' &= x + \Delta x \\
y' &= y + \Delta y \\
\theta' &= \theta + \Delta \theta \\
v'_x &= (x' - x)/\Delta t \\
v'_y &= (y' - y)/\Delta t.
\end{aligned}
\tag{1}
$$

The inverse kinematics can be used to calculate the actions for behavior cloning purpose. It can be described as $\Delta x = x' - x, \Delta y = y' - y, \Delta \theta = \theta' - \theta$.

**Bicycle Action Space**. With the Bicycle action space, we propose a model to approximate the vehicle dynamics with the goal of minimizing the discrepancy between the predicted vehicle states and the recorded vehicle states. More specifically, define the vehicle's coordinates as $x, y$ in the global coordinate system, and the predicted coordinates as $\hat{x}, \hat{y}$, the goal is to minimize $(x - \hat{x})^2 + (y - \hat{y})^2$. Define the current vehicle's state information as $s$, which includes the coordinates of the vehicle in the global coordinate system $(x, y)$, the vehicle's yaw angle $\theta$, the vehicle's speed in the x and y direction $v_x, v_y$. Given the acceleration $a$, and steering curvature $\kappa$, the time length for one step $\Delta t$, the vehicle's next state is calculated using the following *forward dynamics*.

$$
\begin{aligned}
x' &= x + v_x \Delta t + \frac{1}{2} a \cos(\theta) \Delta t^2 \\
y' &= y + v_y \Delta t + \frac{1}{2} a \sin(\theta) \Delta t^2 \\
\theta' &= \theta + \kappa * (\sqrt{v_x^2 + v_y^2} \Delta t + \frac{1}{2} a \Delta t^2) \\
v' &= \sqrt{v_x^2 + v_y^2} + a \Delta t \\
v'_x &= v' \cos(\theta') \\
v'_y &= v' \sin(\theta').
\end{aligned}
\tag{2}
$$

For the inverse kinematics, given the state information of two consecutive states $s = (x, y, \theta, v_x, v_y)$ and $s' = (x', y', \theta', v'_x, v'_y)$, we estimate the acceleration $a$ and steering curvature $\kappa$ using the following equation.

$$
\begin{aligned}
a &= (v' - v)/\Delta t \\
&= (\sqrt{v_x'^2 + v_y'^2} - \sqrt{v_x^2 + v_y^2})/\Delta t \\
\kappa &= (\arctan \frac{v'_x}{v'_y} - \theta)/(\sqrt{v_x^2 + v_y^2} \Delta t + \frac{1}{2} a \Delta t^2).
\end{aligned}
\tag{3}
$$

Using $\arctan \frac{v'_x}{v'_y}$ instead of $\theta'$ empirically achieves smaller prediction error. Other previous environments use a variant of the bicycle model. The steering wheel angle $\theta_{wheel}$ is related with the steering curvature $\kappa$:

$$
\kappa = \frac{\sin(\theta_{wheel}/STEER\_RATIO)}{L},
\tag{4}
$$

where $L$ is the axel length of the vehicle, and $STEER\_RATIO$ is a constant depicting the connection between the front wheel steer angle $\theta_f$ and steering wheel angle $\theta_{wheel}$: $\theta_f = \theta_{wheel}/STEER\_RATIO$.

## A.2 Training Details

**Behavior Cloning Training Details** We re-use the encoder portion of the Wayformer [35] architecture followed by a 4-layer residual MLP (with all hidden layer sizes set to 128) to maximize the log likelihood of the expert actions. For continuous actions, we used a 10-component Gaussian Mixture Model Tanh squashed distribution head. For discrete actions, we used a softmax layer to compute action probabilities. We used Adam with learning rate $1e-4$ and batch size 256.

**DQN Training Details** We used the Acme [19] implementation of prioritized replay double DQN [45]. We used the same architecture as in discrete BC for the Q-network, interpreting the logits of the model as Q-values for each possible action. We used a discount of $\gamma = 0.99$, learning rate $5*10^{-5}$, 1-step Q-learning, a samples-to-insertion ratio of 8, and batch size 64. We trained for 30 million actor steps.

**Hyperparameter Selection** We performed hyperparameter selection for all learned benchmark agents (BC, DQN) outlined in Table 3 via a grid search. For BC, we performed grid search over the learning rate on the values $(3*10^{-5}, 1*10^{-4}, 3*10^{-4})$ and on the action space (Bicycle, Delta, Bicycle-Discrete, Delta-Discrete). For DQN, we performed only grid search over the action space (Bicycle-Discrete, Delta-Discrete).

## A.3 Ablation Study: Runtime and Memory with Number of Objects

We perform an ablation study analyzing the relationship between runtime, memory, and the number of objects simulated. For this ablation study, in the CPU configuration we used a machine with an AMD EPYC 7B12 processor and 64GB RAM. For the GPU configuration we used an Nvidia V100 GPU.

| Device | BS-1 | BS-16 | Objects | Reset | Transition | Metrics | RolloutExpert | Peak Memory |
|--------|------|-------|---------|-------|------------|---------|---------------|-------------|
| CPU | ✓ | | 8 | 0.194 | 0.191 | 0.773 | 121.492 | 5.409 |
| CPU | ✓ | | 16 | 0.184 | 0.176 | 1.431 | 190.357 | 5.590 |
| CPU | ✓ | | 32 | 0.197 | 0.223 | 2.428 | 378.926 | 5.956 |
| CPU | ✓ | | 64 | 0.225 | 0.221 | 4.468 | 637.125 | 6.652 |
| CPU | ✓ | | 128 | 0.286 | 0.274 | 9.831 | 1159.158 | 8.036 |
| CPU | | ✓ | 8 | 1.741 | 2.004 | 10.689 | n/a | 84.066 |
| CPU | | ✓ | 16 | 1.744 | 1.894 | 20.069 | n/a | 86.805 |
| CPU | | ✓ | 32 | 2.084 | 2.414 | 33.002 | n/a | 92.283 |
| CPU | | ✓ | 64 | 2.575 | 2.648 | 66.486 | n/a | 103.239 |
| CPU | | ✓ | 128 | 2.837 | 3.080 | 124.530 | n/a | 125.151 |
| GPU | ✓ | | 8 | 0.250 | 0.265 | 0.159 | 27.010 | - |
| GPU | ✓ | | 16 | 0.253 | 0.267 | 0.158 | 28.041 | - |
| GPU | ✓ | | 32 | 0.258 | 0.268 | 0.208 | 30.488 | - |
| GPU | ✓ | | 64 | 0.260 | 0.276 | 0.157 | 33.206 | - |
| GPU | ✓ | | 128 | 0.246 | 0.257 | 0.152 | 36.856 | - |
| GPU | | ✓ | 8 | 0.264 | 0.266 | 0.154 | n/a | - |
| GPU | | ✓ | 16 | 0.258 | 0.264 | 0.175 | n/a | - |
| GPU | | ✓ | 32 | 0.251 | 0.268 | 0.221 | n/a | - |
| GPU | | ✓ | 64 | 0.280 | 0.289 | 0.301 | n/a | - |
| GPU | | ✓ | 128 | 0.262 | 0.272 | 0.469 | n/a | - |

Table 6: Runtime and memory ablation study over number of objects simulated. All runtimes are reported in milliseconds, and peak memory reported in MB. BS-1 refers to a batch size of 1, and BS-16 refers to a batch size of 16.

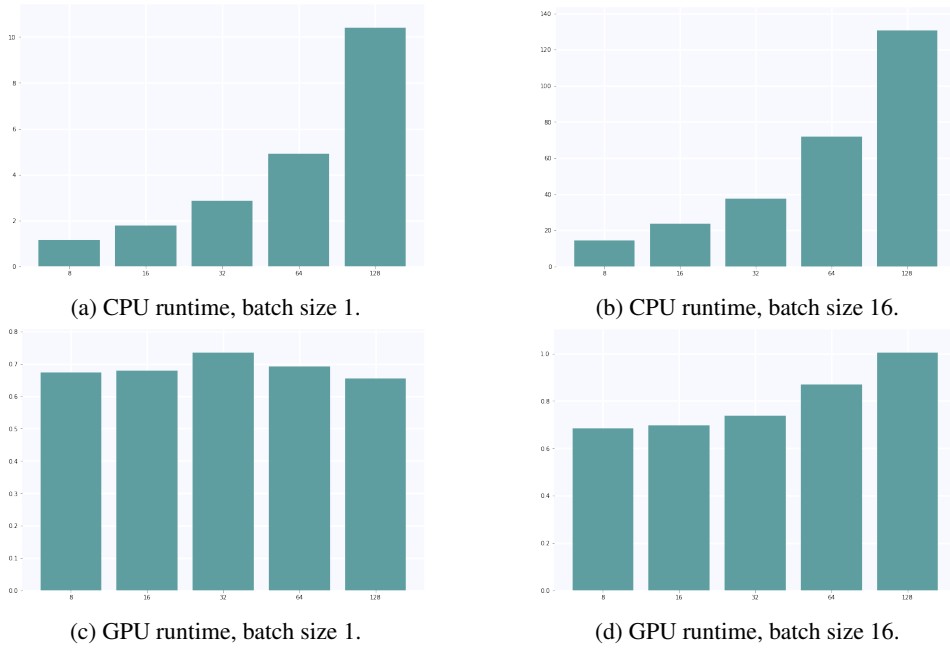

(a) CPU runtime, batch size 1.

(b) CPU runtime, batch size 16.

(c) GPU runtime, batch size 1.

(d) GPU runtime, batch size 16.

Figure 4: Runtime in milliseconds (y-axis) plotted against number of objects simulated (x-axis). The runtime reported is the sum of Reset + Transition + Metrics. Note that while CPU runtime scales linearly with the number of objects simulated, GPU performance is not saturated under the same experimental parameters.

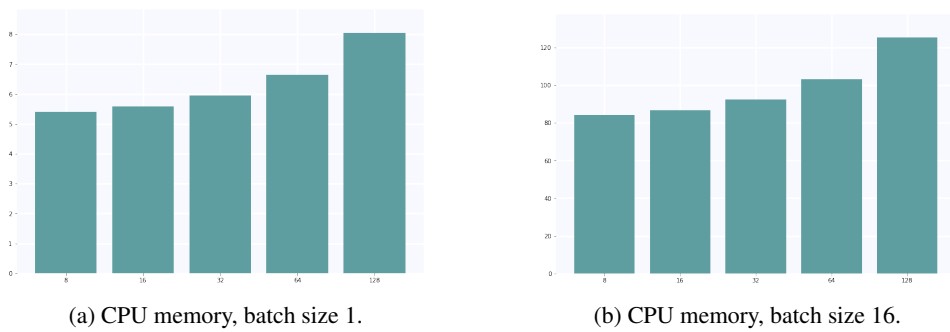

(a) CPU memory, batch size 1.

(b) CPU memory, batch size 16.

Figure 5: Memory usage in megabytes (y-axis) plotted against number of objects simulated (x-axis). The runtime reported is sampled during the execution of the rollout function. Memory usage has a fixed cost then scales roughly linearly with the number of objects