# OpenReview forum: "Waymax: An Accelerated, Data-Driven Simulator for Large-Scale Autonomous Driving Research"
_NeurIPS.cc/2023/Track/Datasets_and_Benchmarks — NeurIPS 2023 Datasets and Benchmarks Poster_

### Official Review · Reviewer_1Xpz · 2023-07-03
**Review of DriveMax: An Accelerated, Data-Driven Simulator for Large-Scale Autonomous Driving Research**

**Rating:** 7
**Confidence:** 2
**Correctness:** The claims made are correct. The benc…
**Clarity:** The paper is clearly written.

**Strengths:**

Although I am not an expert in the area of autonomous driving research, the authors have articulated the need for DriveMax quite well through a comparison to other comparable simulators. It is clear that a large amount of effort has been put towards making this simulator useful for researchers. This includes making it fully differentiable by programming it in JAX, providing both stateful and stateless versions of the main interface, and by designing DriveMax as a set of interoperable libraries, enabling researchers to pick and choose components that are most useful for them. In addition, the authors directly demonstrate the utility of their simulator through several direct benchmarks and ablation studies.

**Additional Feedback:**

N/A

**Documentation:**

The authors note that the training code contains proprietary dependencies and therefore cannot be released. The work is likely reproducible based on the provided information but with some effort. Moreover, the documentation submitted in supplementary materials is extremely detailed and should support good use of the library.

**Ethics:**

No ethical concerns.

**Limitations:**

The authors clearly note that DriveMax does not include any kind of sensor simulation. Moreover, the authors provide an extensive set of benchmarking studies in Section 5 that help to illustrate the potential weaknesses of the model.

**Opportunities For Improvement:**

It seems there might be an opportunity to better situate this work with respect to other simulators. The binary comparison provided in Table 1 is informative, but several of the columns may take on values that are not strictly binary (e.g., integer value for max number of agents rather than simply a binary value for multi-agent).

It looks like Figures 1 and 2b are never referenced in the text. Although they are relevant to the work, there may be an opportunity here to use the space to highlight something that is more germane to DriveMax itself.

The kinematic infeasibility metric seems especially useful, but the it is unclear how the parameters were determined (namely, the 6 m/s2 limit on acceleration magnitude and the 0.3 m-1 limit on steering curvature magnitude). The authors note that these were determined empirically, but some discussion on the procedure would be useful.


**Relation To Prior Work:**

The relationship to prior work is well-articulated with potential for improvement (stated under "Opportunities for Improvement").

**Summary And Contributions:**

This paper presents DriveMax, a multi-agent simulator for research on autonomous driving. It is designed to provide a variety of utilities such as a suite of metrics, a built-in behavior model, and the ability to incorporate real-world datasets. In addition to the simulator itself, the authors provide an extensive series of benchmarks.

---

> ### Author Response · Authors · 2023-08-21
> **Response to Reviewer 1Xpz**
>
> Thank you for the feedback. One of our goals is that DriveMax can be used even for non autonomous driving researchers (for example, as an evaluation environment for RL algorithms) to show impact on a significant real-world problem. We address specific concerns regarding prior work and writing details below:
>
> > “It seems there might be an opportunity to better situate this work with respect to other simulators. The binary comparison provided in Table 1 is informative, but several of the columns may take on values that are not strictly binary (e.g., integer value for max number of agents rather than simply a binary value for multi-agent).”
>
> Thank you for the suggestion. We have updated Table 1 Column “Multi-agent” to reflect the suggested change. Please note that most of the simulators (including DriveMax) do not have an explicit cap on the maximum number of simulated agents and do not report the maximum number of agents. We provide the numbers that are inferred from simulator reports, where typically a figure is shown to illustrate certain aspects of the performance w.r.t. the number of agents - we assume that the simulators at least support the maximum number of agents shown in their respective figures. For those we could not find references, we simply put check marks.
>
> > The kinematic infeasibility metric seems especially useful, but the it is unclear how the parameters were determined (namely, the 6 m/s2 limit on acceleration magnitude and the 0.3 m-1 limit on steering curvature magnitude). The authors note that these were determined empirically, but some discussion on the procedure would be useful.
>
> Thank you for pointing out the need for clarifying information on the kinematic infeasibility metric. By determined empirically, we meant that we analyzed the true distribution of acceleration and steering curvature for planning agents in the WOMD dataset. We then chose the limits to be roughly the maximum (rounding up for some slack) of the values we observed in the logs. We have updated Sec. 3.4 to reflect this.
>
> > It looks like Figures 1 and 2b are never referenced in the text. Although they are relevant to the work, there may be an opportunity here to use the space to highlight something that is more germane to DriveMax itself.
>
> Thank you, we have now added references to Fig 1 and 2b in the text. Our intention with Figure 1 is to show some intuitive examples of what scenarios can be simulated in DriveMax, highlighting some of the specific and beneficial aspects such as multiple simulated-agents and waypoints along the green route. We included this figure because we deemed it important for readers to have a visual understanding of the setting when they initially read the paper. Likewise, Fig 2b is the only demonstration of reactive sim agents that we show in the text. We feel that it is necessary for readers to gain a visual understanding of what their behavior is in simulation because it is a key component of our evaluation. If in addition the reviewer has further suggestions to increase germaneness, we can adapt these images further.

---

> > ### Comment · Reviewer_1Xpz · 2023-08-29
> >
> > Thank you for addressing my comments!

---

### Official Review · Reviewer_ngUf · 2023-07-21
**GPU-based differentiable simulator for close-loop autonomous driving policy training/evaluation**

**Rating:** 8
**Confidence:** 5
**Correctness:** Yes. The results are sound are correct.
**Clarity:** Easy to rea.

**Strengths:**

In recent years, much progress in the robotics field are made with IsaacGym which can run large-scale parallel physics simulations to expedite data sampling and policy training. I am glad to see that the autonomous driving community finally has our GPU-based simulator now. It can facilitate traversing large-scale real-world data quickly and would definitely benefit ML-based planner research. In addition, the differentiable system dynamics open new research opportunities for trajectory prediction and policy learning. For example, the policy can be optimized without estimating the value function as the derivate of the reward function can serve as a gradient for updating the policy network.
The experiment section proves that the simulator is ready for being a new testbed where users can develop/test new data-driven planning algorithms and enjoy the efficiency.

**Additional Feedback:**

In Table 3, the gap between *Log* and *ADE* can be reduced. Or just place *ADE* to the next line.

I think it can make the table more clear. (It is fine to ignore this advice as it is my personal preference)

**Documentation:**

The supplementary provides all code and docs.

**Limitations:**

Limitations should be discussed as self-driving is a safety-critical application. This could be emphasizing that the policy trained in the simulator can not be deployed on a real car due to the sim2real gap.

**Opportunities For Improvement:**

I enjoy reading this paper. Every part is well-organized and easy to understand. The only suggestion I have is to add an experiment to show how the differentiable simulation can benefit policy learning. It could be training a policy gradient baseline which is optimized by differentiating the reward function directly. A good reference for this is:

```
PODS: Policy Optimization via Differentiable Simulation
Miguel Angel Zamora Mora, Momchil Peychev, Sehoon Ha, Martin Vechev, Stelian Coros
International Conference on Machine Learning (ICML) 2021
```
I just feel that it would be better to have an experiment to highlight this feature or at least demonstrate that it is differentiable empirically.

Typo: line 258: then->than

**Relation To Prior Work:**

As far as I know, it covers almost every important work prior to this.

**Summary And Contributions:**

For developing/validating planners taking mid-level representation as input, it is important to guarantee speed and realism. All driving simulators run on CPU and most of them focus on hand-crafted scenarios. Thus the authors made a new simulator, DriveMax, to address both issues. The new simulator reconstructs real-world scenarios and rolls out the simulated world on GPU/TPU, which provides guarantees for realism and efficiency. In the experiment section, the runtime benchmark is provided to prove the efficiency. In addition, several baseline policy are trained to prove the effectiveness of DriveMax for ML-based planner learning.

---

> ### Author Response · Authors · 2023-08-21
> **Response to Reviewer ngUf**
>
> Thank you for the feedback. We address specific concerns regarding baselines and limitations below:
>
> > Limitations should be discussed as self-driving is a safety-critical application. This could be emphasizing that the policy trained in the simulator can not be deployed on a real car due to the sim2real gap.
>
> The sim2real is gap is a critical problem in self-driving as the reviewer mentions. We have expanded discussion on this limitation in the conclusion, including now referencing additional works in the area of sim2real such as [1], [2], [3] (we also welcome additional references on this topic). While there is no easy solution to this problem for any simulator, we believe that the DriveMax software is modular and adaptable enough that advanced sim2real methods can be implemented and explored within the simulator.
>
> As DriveMax is primarily intended to be a research tool and not a production-level simulator, we will include language in the license for DriveMax that prevents the usage of models trained in DriveMax in operation of real cars, as a mitigation for the potential safety risks of deploying models in real-life scenarios.
>
> [1] Tobin, Josh, et al. "Domain randomization for transferring deep neural networks from simulation to the real world." 2017 IEEE/RSJ international conference on intelligent robots and systems (IROS). IEEE, 2017
> [2] Osiński, Błażej, et al. "Simulation-based reinforcement learning for real-world autonomous driving." 2020 IEEE international conference on robotics and automation (ICRA). IEEE, 2020.
> [3] Herzog, Alexander, et al. "Deep RL at Scale: Sorting Waste in Office Buildings with a Fleet of Mobile Manipulators." arXiv preprint arXiv:2305.03270 (2023).
>
> > “The only suggestion I have is to add an experiment to show how the differentiable simulation can benefit policy learning. It could be training a policy gradient baseline which is optimized by differentiating the reward function directly”
>
> We agree that such an experiment would be a useful addition. Existing work in the self-driving space has shown the benefits of differentiable simulation for policy learning (e.g., [4]), albeit with a proprietary simulator. We did not have sufficient time to add this study, but we have added a discussion of this as future work and cited [5]. We hope that our work enables researchers to conduct this follow up.
>
> [4] Igl, Maximilian, et al. "Symphony: Learning realistic and diverse agents for autonomous driving simulation." 2022 International Conference on Robotics and Automation (ICRA). IEEE, 2022.
>
> [5] Miguel Angel Zamora Mora, Momchil Peychev, Sehoon Ha, Martin Vechev, Stelian Coros. “PODS: Policy Optimization via Differentiable Simulation” International Conference on Machine Learning (ICML) 2021.

---

### Official Review · Reviewer_DvxP · 2023-07-24
**The paper presents DriveMax, a simulator for autonomous driving designed for broad planning research. It provides a computational evaluation, an empirical study of several benchmark agents, and benchmarks various planning algorithms.**

**Rating:** 6
**Confidence:** 4
**Clarity:** Yes

**Strengths:**

The main strength of this submission is the introduction of DriveMax, a versatile and efficient autonomous driving simulator that allows various configurations of the learning agent and the environment. This contributes significantly to the field of autonomous driving research by providing a tool for researchers to design and evaluate different learning and planning algorithms in a controllable and reproducible environment. The research appears to be of high quality, as evidenced by comprehensive benchmarks and thoughtful experimental designs. The simulator, combined with detailed benchmarks and baseline planning agents, can serve as a valuable resource for the research and development of safer and more efficient self-driving technologies.

**Additional Feedback:**

Please provide the necessary details missing in the Documentation section.

**Correctness:**

It is a benchmark. But I am not sure if the authors plan to just release the code for DriveMax or will also set up a benchmarking server like nuPlan.

**Documentation:**

- I didn't find any documentation regarding organization and maintenance, and ethical and responsible use.
- There was no maintenance plan provided.
- I am not sure if the detail available is sufficient enough to support reproducibility. Also, I might have missed it, but in the documentation provided, I didn't find anything for evaluation.


**Limitations:**

Yes

**Opportunities For Improvement:**

- 'DriveMax is a simulator that supports controlling an arbitrary number of objects in a scene' - Given the computation overhead with each object, how many large-size objects can be controlled? A graph with memory vs the number of objects would give a better understanding of the limits. (I reckon it can't go higher than WOMD)


**Relation To Prior Work:**

Yes

**Summary And Contributions:**

The authors introduce DriveMax, a novel simulator for autonomous driving aimed at facilitating extensive planning research. They demonstrated the computation performance of DriveMax under various configurations, showing its ability to efficiently handle single-agent and multi-agent environments. The paper introduced various baseline planning agents and showcased how they operate in the DriveMax environment. Further, the authors performed several ablation studies to highlight the impact of route conditioning and simulation agent type on the performance of trained models.

---

> ### Author Response · Authors · 2023-08-21
> **Response to Reviewer DvxP**
>
> Reviewer #1
>
> Thank you for your insightful comments. We are excited to release a simulation platform that is realistic, differentiable, and computationally efficient, and welcome as much feedback as possible to improve the simulator. We address specific concerns regarding performance, maintenance, and reproducibility below:
>
> > I am not sure if the detail available is sufficient enough to support reproducibility. Also, I might have missed it, but in the documentation provided, I didn't find anything for evaluation.
>
> We are unfortunately unable to provide code for our baselines as it contains proprietary information for our organization, but an open-source version of the training code we used is available on github at https://github.com/deepmind/acme. The work required to reproduce our experimental setup would be to implement the network architectures described in Appendix A.2 and running the appropriate algorithm (e.g. BC, DQN) in ACME.
>
> For evaluation we defined the set of metrics we considered in Section 3.4 and shared our results in Table 3. Furthermore, the evaluation metrics will be publicly available in the Drivemax source code when released allowing users clear understanding of their implementation. If you have any further clarifying questions on evaluation, please let us know so we can improve the camera ready.
>
> > 'DriveMax is a simulator that supports controlling an arbitrary number of objects in a scene' - Given the computation overhead with each object, how many large-size objects can be controlled? A graph with memory vs the number of objects would give a better understanding of the limits. (I reckon it can't go higher than WOMD)
>
> We have added a section analyzing the runtime and memory usage vs the maximum number of objects simulated in Appendix A.3. The current limit is 128 objects per scenario, but only because that is the maximum number of objects contained in WOMD, and the actual maximum number of simulated objects in memory depends on other factors such as the simulation batch size (number of simulations to run in parallel). Factoring batch size, we are simulating 2048 objects (128 agents * 16 batch size) simultaneously in our experiments.
>
> We show now in Appendix A.3 that both memory and runtime scales roughly linearly on CPU with the number of objects and batch size, with a constant overhead. GPU scaling tends to be sublinear until the GPU is fully saturated, upon which it will also tend to linear scaling. The main memory cost is due to roadgraph operations (since the roadgraph in WOMD consists of 20k points) - determining whether each object is offroad will require operations with this large data structure.
>
> > I didn't find any documentation regarding organization and maintenance, and ethical and responsible use. There was no maintenance plan provided.
>
> We thank the reviewer for bringing to light the important considerations for the maintenance of the Drivemax code. Our code will be hosted on Github following the camera ready release and we will allow update proposals via a Github issue and are open to any suggestions to make the DriveMax codebase better. We are committed to the code maintenance and will respond to user requests, via email and Github issues for at least 12 months after publishing the code on Github.
>
> > It is a benchmark. But I am not sure if the authors plan to just release the code for DriveMax or will also set up a benchmarking server like nuPlan.
>
> Our target is to have this publicly announced benchmarking server and a public challenge next year to facilitate evaluation and benchmarking, pending necessary internal approvals.

---

> > ### Comment · Reviewer_DvxP · 2023-08-30
> >
> > I thank the authors for the response. I will keep my rating.

---

### Decision · Program_Chairs · 2023-09-22

**Decision:**

Accept (Poster)

**Comment:**

This paper presents a novel simulator, DriveMax, for autonomous driving aimed at facilitating extensive planning tasks. The research appears to be of high quality to the whole autonomous driving field. All reviewers give positive comments. Extensive studies are provided.The area chair thus decides to accept it.